# Beyond Smoothness: Incorporating Low-Rank Analysis into Nonparametric Density Estimation

**Robert A. Vandermeulen**
Machine Learning Group
Technische Universiät Berlin
Berlin, Germany
vandermeulen@tu-berlin.de

**Antoine Ledent**
Machine Learning Group
Technische Universität Kaiserslautern
Kaiserslautern, Germany
ledent@cs.uni-kl.de

## Abstract

The construction and theoretical analysis of the most popular universally consistent nonparametric density estimators hinge on one functional property: smoothness. In this paper we investigate the theoretical implications of incorporating a multi-view latent variable model, a type of low-rank model, into nonparametric density estimation. To do this we perform extensive analysis on histogram-style estimators that integrate a multi-view model. Our analysis culminates in showing that there exists a universally consistent histogram-style estimator that converges to any multi-view model with a finite number of Lipschitz continuous components at a rate of $\widetilde{O}(1/\sqrt[3]{n})$ in $L^1$ error. In contrast, the standard histogram estimator can converge at a rate slower than $1/\sqrt[d]{n}$ on the same class of densities. We also introduce a new nonparametric latent variable model based on the Tucker decomposition. A rudimentary implementation of our estimators experimentally demonstrates a considerable performance improvement over the standard histogram estimator. We also provide a thorough analysis of the sample complexity of our Tucker decomposition-based model and a variety of other results. Thus, our paper provides solid theoretical foundations for extending low-rank techniques to the nonparametric setting.

## 1 Introduction

Nonparametric density estimators are density estimators capable of estimating a density $p$ while making few to no assumptions on $p$. Two commonly used nonparametric density estimators include the histogram estimator and the kernel density estimator (KDE). A common characteristic of these two estimators is that they make estimation tractable via a hyperparameter that relates to smoothness, namely bin width and bandwidth. Large bin width or bandwidth allows for good control of estimation error[*] at the cost of increased approximation error via large bin volume for the histogram and smoothing for KDEs. Selection of this parameter is crucial for estimator performance. In fact a recent survey on bandwidth selection for KDEs found at least 30 methods for setting this value along with a few surveys dedicated to the topic [15].

While nonparametric density estimation has been shown to be effective for many tasks, it has been observed empirically that estimator performance typically declines as data dimensionality increases, a manifestation of the curse of dimensionality. For the histogram and KDE this phenomenon has concrete mathematical analogs. For example, these estimators are only universally consistent[†] if

---

[*]For an estimator $V$ restricted space a of densities $\mathcal{P}$, the *estimation error* refers to the difference between $\|V - p\|$ and $\min_{q \in \mathcal{P}} \|p - q\|$, where $p$ is the target density. This is similar to estimator variance.

[†]A density estimator is *universally consistent* if it asymptotically recovers *any* density.

35th Conference on Neural Information Processing Systems (NeurIPS 2021).

$n \to \infty$ and $h \to 0$, with $nh^d \to \infty$, where $n$ is the number of samples, $d$ is the data dimension, and $h$ is the bin width for the histogram and bandwidth parameter for the KDE [14]. One may then wonder whether there exists some other way to constrain model capacity so as to alleviate this exponential dependence on dimension.

In this paper we theoretically analyze the advantages of including a constraint akin to matrix/tensor rank in a nonparametric density estimator. We do so by analyzing histogram-style estimators (estimators that output a density that are piecewise constant on bins defined by a grid) that are enforced to have a low-rank PARAFAC or Tucker decomposition. Enforcing this low-rank constraint in the histogram estimator allows for much faster rates on $h \to 0$, while still controlling estimation error. This can remove the exponential penalty on rate of convergence that dimensionality produces with the standard histogram estimator.

The bulk of this work focuses on analysis of hypothetical estimators that, while offering large statistical advantages over the standard histogram estimator, are not computationally tractable. At the end of this work we include experiments demonstrating that low-rank histograms, computed using an off-the-shelf nonnegative tensor factorization library [23], consistently outperform the standard histogram estimator.

## 1.1 Density Models

Here we introduce the low-rank models that will be used to constrain our estimators. The first is a *multi-view model*. A multi-view model $p$ is a finite mixture $p = \sum_{i=1}^{k} w_i f_i$ whose components are separable densities $f_i = p_{i,1} \otimes p_{i,2} \otimes \cdots \otimes p_{i,d}$ [‡] [19, 1, 37, 36]. A multi-view model has the following form

$$p(x_1, x_2, \ldots, x_d) = \sum_{i=1}^{k} w_i p_{i,1}(x_1) p_{i,2}(x_2) \cdots p_{i,d}(x_d). \tag{1}$$

A multi-view model with one component is typically called a *naive Bayes model*. For the estimators we propose, the component marginals $p_{i,j}$ will have the form of one-dimensional histograms and the number of components $k$ will be limited to restrict estimator capacity. When $k = 1$ the model is equivalent to a naive Bayes model, $p(x_1, x_2, \ldots, x_d) = p_1(x_1) p_2(x_2) \cdots p_d(x_d)$. When the component marginals $p_{i,j}$ are histograms increasing $k$ expands the set of potential estimates from naive Bayes models when $k = 1$ to all possible histogram estimates.

The models in this paper are motivated by nonnegative tensor factorizations so we term them generally as *nonparametric nonnegative tensor factorization* (NNTF) *models*. The previous model was related to nonnegative PARAFAC [34]. Our second model is based on the nonnegative Tucker decomposition [22]. A density in this model utilizes $d$ collections, $\mathcal{F}_1, \ldots, \mathcal{F}_d$, of $k$ one-dimensional densities, $\mathcal{F}_i = \{p_{i,1}, \ldots, p_{i,k}\}$, and some probability measure that randomly selects one density from each $\mathcal{F}_i$. This measure a can be represented by a tensor $W \in \mathbb{R}^{k \times d}$ where the probability of selecting $(p_{1,i_1}, \ldots, p_{d,i_d})$ from $\mathcal{F}_1 \times \cdots \times \mathcal{F}_d$ is $W_{i_1, \ldots, i_d}$. To sample from this model we first randomly select the marginal distributions $p_{1,i_1}, \ldots, p_{d,i_d}$ according to $W$, and an observation is sampled according to the $d$-dimensional distribution $p_{1,i_1} \otimes p_{2,i_2} \otimes \cdots \otimes p_{2,i_2}$. The density of this model is

$$p(x_1, x_2, \ldots, x_d) = \sum_{i_1=1}^{k} \cdots \sum_{i_d=1}^{k} W_{i_1, \ldots, i_d} p_{1,i_1}(x_1) p_{2,i_2}(x_2) \cdots p_{d,i_d}(x_d). \tag{2}$$

We are unaware of previous works investigating this model for general probability distributions so we will simply term it the *Tucker model*. Again we will investigate estimators where the component marginals are one-dimensional histograms and small $k$ corresponds to reduced model capacity. We remark that Tucker decompositions typically have a rank *vector* $[k_1, \ldots, k_d]$ which for our model would mean that each $\mathcal{F}_i$ contains $k_i$ vectors and $W$ would lie in $\mathbb{R}^{k_1 \times \cdots \times k_d}$. This sort of rank restriction could be used in our methods however, for simplicity, we just set $k_1 = k_2 = \cdots = k_d \triangleq k$.

---

[‡] For two functions $f, g$ let $f \otimes g : (x, y) \mapsto f(x)g(y)$. This is analogous to the tensor product of $L^2$ functions.

## 1.2 Overview of Results

The principal contribution of this paper is to analyze the advantage of incorporating rank restriction into density estimation. In Section 2.1 we precisely introduce the multi-view histogram and a histogram based on the Tucker model. With these models we investigate how quickly we can let bin width $h$ go to zero and rank $k$ grow, in relation to the amount of data $n$, while still being able to control estimation error and select an estimator that is nearly optimal for the allowable set. For the multi-view histogram we show that one can control estimation error so long as $k/h$ is asymptotically dominated by $n$ and for the Tucker histogram we need $k/h + k^d$ to be asymptotically dominated by $n$ (we are omitting logarithmic factors here for convenience). This stands in stark contrast to standard space of histogram estimators which requires $1/h^d$ to be asymptotically dominated by $n$. We furthermore show that these estimators are universally consistent and that these rates cannot be significantly improved.

For a second style of analysis we provide finite-sample bounds on the convergence of NNTF histogram estimators. We then construct a class of universally consistent density estimators that converge at rate $\widetilde{O}(1/\sqrt[3]{n})$[§] on all densities that are a multi-view model with Lipschitz continuous component marginals. Note that the NNTF histogram estimators we construct do not require the target density to be an NNTF model to function well. Our estimators will select a good estimator so long as there exists any NNTF histogram that approximates the target density well; so our hypothetical estimators "fail elegantly" in some sense. We further show that the standard histogram can converge at a rate $\omega(1/\sqrt[d]{n})$[¶] on the same class of densities. In Section 3 we experimentally investigate the efficacy of using NNTF histograms on real-world data. In lieu of the computationally intractable methods we investigated in our theoretical analyses, we use an existing low-rank nonnegative Tucker factorization algorithm to fit an NNTF histogram to data. Surprisingly even this method outperforms the standard histogram estimator with very high statistical significance. Lastly we mention that this paper is an extension of [42] and contains a fair amount of overlap with that text.

## 1.3 Previous Work

Nonparametric density estimation has been extensively studied with the histogram estimator and KDE being some of the most well-known methods. There do exist, however, alternative methods for density estimation, e.g. the *forest density estimator* [24] and *k-nearest neighbor density estimator* [25]. The $L^1$, $L^2$, and $L^\infty$ convergence of the histogram and KDE has been studied extensively [14, 9, 40, 16]. The KDE is generally regarded as the superior density estimator, with some mathematical justification [14, 35]. Numerous modifications and extensions of the KDE have been proposed including utilizing variable bandwidth [39], robust KDEs [20, 41, 44], methods for enforcing support boundary constraints [33], and a supervised variant [43]. The work [46] investigated using KDEs for nonparametric mixture modeling. Finally we mention [21] that demonstrated that uniform convergence of a KDE to its population estimate suffered when the intrinsic dimension of the data was lower than the ambient dimension, a phenomenon seemingly at odds with the curse of dimensionality.

For our review of NNTF models we also include a general review of tensor/matrix factorizations since both can be viewed being low-rank models. In particular, for the multi-view model we have the following analogy

$$\sum_{i=1}^k w_i p_{i,1}(x_1) p_{i,2}(x_2) \cdots p_{i,d}(x_d) \sim \sum_{i=1}^k \lambda_i \mathbf{v}_{i,1} \otimes \mathbf{v}_{i,2} \otimes \cdots \otimes \mathbf{v}_{i,d}. \tag{3}$$

A great deal of work has gone into leveraging low-rank assumptions to improve matrix estimation, particularly in the field of *compressed sensing* [10, 30]. The most basic version of compressed sensing is concerned with estimating a "tall" vector $x$ from a "short" vector $y \triangleq Ax$ where $A$ is a known "short and fat" matrix. One can recover $y$ if it is sparse and $A$ satisfies a property known as the restricted isometry property (RIP) [47, 7]. These methods can be extended to the estimation of matrices when $x$ is a low-rank matrix [30, 26, 27]. In this extension $A$ is an order-3 tensor which acts

---

[§]$f_n \in \widetilde{O}(g_n) \iff \exists k$ s.t. $f_n \in O\left(g_n \log^k g_n\right)$

[¶]$f_n \in \omega(g_n) \iff |f_n/g_n| \to \infty$ in probability. In $d$-dimensional space the standard histogram can converge slower than $1/\sqrt[d]{n}$.

as a linear operator on $x$ and satisfies an adjusted form of RIP. RIP commonly arises from matrices and tensors whose entries are random. Because of this compressed sensing techniques are useful in settings where one wants to estimate $x$ from random linear transforms of $x$. For example, in matrix completion observing the $(i, j)$-th entry of $x$ can be represented as an inner product of $x$ with an indicator matrix $\mathbf{e}_{i,j}$, i.e. $\langle x, \mathbf{e}_{i,j} \rangle_F$. Thus the random observed indices $(i_1, j_1), (i_2, j_2), \ldots$ can be represented as random matrices $\mathbf{e}_{i_1, j_1}, \mathbf{e}_{i_2, j_2}, \ldots$ which are then stacked into $A$. Now $A$ is a random linear operator that is applied to $x$ to represent the observation of random entries of $x$ and the methods of compressed sensing can be used to recover $x$. Compressed sensing has also proven useful for multivariate regression and autoregressive models [26, 27]. Such techniques don't appear to be extensible to histogram estimation due to the lack of a linear sampling scheme.

General matrix/tensor factorization, including nonnegative matrix/tensor factorizations, has been extensively studied despite being inherently difficult due to non-convexity. The identifiability and recovery of mixtures of product distributions was studied in [18, 38, 13]. The special case of mixtures of power distributions has also been investigated [29, 45]. The works [11, 5] present potential theoretical grounds for avoiding the computational difficulties of nonnegative matrix factorization. One notable approach to tensor factorization is to assume, in the tensor representation in (3), that $d \geq 3$ and the collections of vectors $\mathbf{v}_{1,j}, \ldots, \mathbf{v}_{k,j}$ are linearly independent for all $j$. Under this assumption we are guaranteed that the factorization (3) is unique [1, 18]. In [4] the authors present a method for recovering this factorization efficiently and demonstrate its utility for a variety of tasks. This work was extended in [36] to recover a multi-view KDE satisfying an analogous linear independence assumption and theoretically analyze the estimator's convergence to the true low-rank components. In [36] the authors investigate the sample complexity of their estimator but do not demonstrate that their technique has potential for improving rates for nonparametric density estimation in general. In [37] it was observed that using low-rank embeddings can improve density estimation. A multi-view histogram was investigated in [17] where the authors present an identifiability result and algorithm for recovering latent factors of the distribution. The related works [2, 3] consider a low-rank characteristic function as an approach to improving nonparametric density estimation. Though earlier works have observed that a low-rank approach improves nonparametric density estimation [37, 36, 17, 2, 3], we are the first to demonstrate this through theoretical analysis of sample complexity. Finally we note that the Tucker decomposition has been utilized in Bayesian statistics [32]. We are unaware of any literature on factoring functions in $\mathbb{R}^d \to \mathbb{R}$ in a Tucker-inspired as we do in (2).

## 2   Theoretical Results

In this section we mathematically demonstrate that histogram estimators can achieve greater performance by restricting to NNTF models and using a proper procedure to select a representative from these using data. To simplify analysis we will only consider densities on the unit cube $[0, 1)^d$ and analyze the number of bins per dimension $b$ which is the inverse of the bin width, i.e. $b = 1/h$. To state our results precisely we must introduce a fair amount of notation.

### 2.1   Notation

We will denote the $L^1$ and $L^2$ Lebesgue space norms via a 1 or 2 subscript. Let $\mathcal{D}_d$ be the set of all densities on $[0, 1)^d$. By *density* we mean probability measures that are absolutely continuous with respect to the $d$-dimensional Lebesgue measure on $[0, 1)^d$. We define a *probability vector* or *probability tensor* to simply mean a vector or tensor whose entries are nonnegative and sum to one. Let $\Delta_b$ denote the set of probability vectors in $\mathbb{R}^b$ and $\mathcal{T}_{d,b}$ the set of probability tensors in $\mathbb{R}^{b \times d}$ [‖]. The product symbol $\prod$ will always mean the standard outer product, e.g. set product[**] or tensor product, when the multiplicands are not real numbers[††]. The natural numbers $\mathbb{N}$ will always denote *positive* integers. For any natural number $b$ let $[b] = \{1, \ldots, b\}$. We will let $\mathbb{1}$ be the indicator function and $\mathrm{Conv}$ be the convex hull. Later we will use projection operator where $\mathrm{Proj}_S x \triangleq \arg\min_{s \in S} \|x - s\|_2$; for every instance in this work this projection is unique.

---

[‖] $\mathbb{R}^{b \times d}$ is the set of $\underbrace{b \times \cdots \times b}_{d \text{ times}}$ tensors. For example $\mathbb{R}^{b \times 2}$ is the set of $b \times b$ matrices.

[**] For sets $S_1, \ldots, S_d$ we have $\prod_{i=1}^d S_i = S_1 \times \cdots \times S_d = \{(s_1, \ldots, s_d) : s_i \in S_i \forall i\}$.

[††] For functions $f_1, \ldots, f_d$ then $\prod_{i=1}^d f_i = f_1 \otimes \cdots \otimes f_d : (x_1, \ldots, x_d) \mapsto \prod_{i=1}^d f_i(x_i)$.

We will now construct the space of histograms on $[0, 1)^d$. We begin with one-dimensional histograms, which will serve as the $p_{i,j}$ terms in (1) or (2). We define $h_{1,b,i}$ with $i \in [b]$ to be the one-dimensional histogram where all weight is allocated to the $i$th bin. Formally we define this as

$$h_{1,b,i}(x) \triangleq b \mathbb{1}\left( \frac{i-1}{b} \leq x < \frac{i}{b} \right).$$

Note that this is a valid density due to the leading $b$ coefficient. We use these to construct higher-dimensional histograms. For a multi-index $A \in [b]^d$, let

$$h_{d,b,A} \triangleq \prod_{i=1}^{d} h_{1,b,A_i},$$

i.e. the $d$-dimensional histogram with $b$ bins per dimension whose entire density is allocated to the bin indexed by $A$. Finally we define $\Lambda_{d,b,A}$ to be the support of $h_{d,b,A}$, i.e. the "bins" of a histogram estimator,

$$\Lambda_{d,b,A} \triangleq \prod_{i=1}^{d} \left[ \frac{A_i - 1}{b}, \frac{A_i}{b} \right).$$

For a dataset $\mathcal{X} = (X_1, \ldots, X_n)$ in $[0, 1)^d$, the standard histogram estimator is

$$H_{d,b}(\mathcal{X}) \triangleq \frac{1}{n} \sum_{i=1}^{n} \sum_{A \in [b]^d} h_{d,b,A} \mathbb{1}(X_i \in \Lambda_{d,b,A}).$$

Let $\mathcal{H}_{d,b} \triangleq \text{Conv}\left( \left\{ h_{d,b,A} \,\middle|\, A \in [b]^d \right\} \right)$, the set of all $d$-dimensional histograms with $b$ bins per dimension. Let $\mathcal{H}_{d,b}^k$ be the set of histograms with at most $k$ separable components, i.e.

$$\mathcal{H}_{d,b}^k \triangleq \left\{ \sum_{i=1}^{k} w_i \prod_{j=1}^{d} p_{i,j} \,\middle|\, w \in \Delta_k, p_{i,j} \in \mathcal{H}_{1,b} \right\}. \tag{4}$$

We will refer to elements in this space as *multi-view histograms*. Elements in this space have the same form as (1) in Section 1.1. Similarly we define the space of *Tucker histograms* to be

$$\widetilde{\mathcal{H}}_{d,b}^k = \left\{ \sum_{S \in [k]^d} W_S \prod_{i=1}^{d} p_{i,S_i} \,\middle|\, W \in \mathcal{T}_{d,k}, p_{i,j} \in \mathcal{H}_{1,b} \right\}.$$

These have the same form as (2) in Section 1.1.

We emphasize that the collections of densities $\mathcal{H}_{d,b}^k$ and $\widetilde{\mathcal{H}}_{d,b}^k$ are primary objects of interest in this paper. The results we present are concerned with finding good density estimators restricted to these sets as $k$ and $b$ vary.

## 2.2 Estimator Theoretical Results

We present two approaches to the analysis of NNTF histogram estimators. All proofs and additional results are contained in the appendix.

First we provide an asymptotic analysis of NNTF histogram estimators in terms of estimation error control: how fast can we let $b$ and $k$ grow, with respect to $n$, while still controlling estimation error over all densities? An advantage of this analysis is that we can demonstrate that these rates are approximately optimal (up to logarithmic terms).

For our second approach we present finite-sample bound analysis. In this analysis we first present a distribution-dependent bound that, for an estimator restricted to $\mathcal{H}_{d,b}^k$, depends on $n, b, k$ and $\min_{q \in \mathcal{H}_{d,b}^k} \|p - q\|_1$ where $p$ is the data generating "target" distribution. We follow this up with distribution-free bounds.

Distribution-free bounds for nonparametric density estimation require that the target distribution belong to a well-behaved class of distributions (such as Sobolev or Hölder classes) to enable bounding of the approximation error [40]. Our distribution-free finite-sample analysis assumes that the target density is a multi-view model whose component marginals are Lipschitz continuous. We construct an estimator that converges at a rate of approximately $1/\sqrt[3]{n}$ on this class of densities, independent of $d$. For comparison we show that the standard histogram estimator can converge at a rate worse than $1/\sqrt[d]{n}$ on this same class of densities. We mention again that this estimator shouldn't fail catastrophically when these distributional assumptions aren't exactly met so long as its approximation error, $\min_{p \in \mathcal{H}_{d,b}^k} \|p - q\|_1$, isn't large. For brevity, and because the results are virtually direct analogues of their multi-view histogram counterparts, we reserve all finite-sample results for Tucker histogram for the appendix.

**Main Technical Tools**  Our results rely on finding $L^1$ $\varepsilon$-coverings of the spaces of NNTF histograms and using Theorem 3.7 from [6] to select a good representative from that collection. The aforementioned theorem is a slight extension of Theorem 6.3 in [9] and is essentially proven in [48]. As was mentioned in [6], the application of these results typically does not yield a computationally practical algorithm. Likewise our results are simply meant to highlight the potential of NNTF models and are not practically implementable as is.

### 2.2.1  Asymptotic Error Control

The following theorem states that one can control the estimation error of multi-view histograms with $k$ components and $b$ bins per dimension so long as $n \sim bk$ (omitting logarithmic factors). Recall that the standard histogram estimator requires $n \sim b^d$, so we have removed the exponential dependence of bin rate on dimensionality. Here and elsewhere the $\sim$ symbol is not a precise mathematical statement but rather signifies that the two values should be of the same order in a general sense. In the following $b$ and $k$ are functions of $n$ so the space of histograms changes as one acquires more data.

**Theorem 2.1.** *For any pairs of sequences $b \to \infty$ and $k \to \infty$ satisfying*
$$n/(bk \log(b) + k \log(k)) \to \infty,$$
*there exists an estimator $V_n \in \mathcal{H}_{d,b}^k$ such that, for all $\varepsilon > 0$*
$$\sup_{p \in \mathcal{D}_d} P\left( \|V_n - p\|_1 > 3 \min_{q \in \mathcal{H}_{d,b}^k} \|p - q\|_1 + \varepsilon \right) \to 0,$$
*where $V_n$ is a function of $X_1, \ldots, X_n \overset{iid}{\sim} p$.*

The sample complexity for the multi-view histogram is perhaps more accurately approximated as being on the order of $dbk$ however the $d$ disappears in the asymptotic analysis. The following theorem states that one can control the error of Tucker histogram estimates so long as $n \sim bk + k^d$ (omitting logarithmic factors).

**Theorem 2.2.** *For any pairs of sequences $b \to \infty$ and $k \to \infty$ satisfying*
$$n/\left( bk \log(b) + k^d \log\left(k^d\right) \right) \to \infty,$$
*there exists an estimator $V_n \in \widetilde{\mathcal{H}}_{d,b}^k$ such that, for all $\varepsilon > 0$*
$$\sup_{p \in \mathcal{D}_d} P\left( \|V_n - p\|_1 > 3 \min_{q \in \widetilde{\mathcal{H}}_{d,b}^k} \|p - q\|_1 + \varepsilon \right) \to 0,$$
*where $V_n$ is a function of $X_1, \ldots, X_n \overset{iid}{\sim} p$.*

Allowing $b$ to grow as aggressively as possible we achieve consistent estimation so long as $n \sim b \log b$ and $k$ grows sufficiently slowly, regardless of dimensionality.

**Corollary 2.1.** *For all $d, b, k$ fix $\mathcal{R}_{d,b}^k$ to be either $\mathcal{H}_{d,b}^k$ or $\widetilde{\mathcal{H}}_{d,b}^k$ [‡‡]. For any sequence $b \to \infty$ with $n/\left( b \log b \right) \to \infty$, there exists a sequence $k \to \infty$ and estimator $V_n \in \mathcal{R}_{d,b}^k$ such that, for all $\varepsilon > 0$*
$$\sup_{p \in \mathcal{D}_d} P\left( \|V_n - p\|_1 > 3 \min_{q \in \mathcal{R}_{d,b}^k} \|p - q\|_1 + \varepsilon \right) \to 0,$$

---

[‡‡]$\mathcal{R}$ is fixed to $\mathcal{H}$ or $\widetilde{\mathcal{H}}$ and doesn't change as $n, b, k$ vary.

where $V_n$ is a function of $X_1, \ldots, X_n \overset{iid}{\sim} p$.

The following result shows that the approximation error of the estimators in Theorem 2.1, Theorem 2.2, and Corollary 2.1 go to zero for all densities. Thus these estimators are universally consistent even when the NNTF model assumption is not satisfied.

**Lemma 2.1.** *Let* $p \in \mathcal{D}_d$. *If* $k \to \infty$ *and* $b \to \infty$ *then* $\min_{q \in \mathcal{H}_{d,b}^k} \|p - q\|_1 \to 0$.

A straightforward consequence of this is that the Tucker histogram approximation error also goes to zero.

**Lemma 2.2.** *Let* $p \in \mathcal{D}_d$. *If* $k \to \infty$ *and* $b \to \infty$ *then* $\min_{q \in \widetilde{\mathcal{H}}_{d,b}^k} \|p - q\|_1 \to 0$.

The next theorem shows that the rate on $bk$ in Theorem 2.1 cannot be made significantly faster.

**Theorem 2.3.** *Let* $d \geq 2$, $b \to \infty$, *and* $k \to \infty$ *with* $b \geq k$ *and* $n/(bk) \to 0$. *There exists no estimator* $V_n \in \mathcal{H}_{d,b}^k$ *such that, for all* $\varepsilon > 0$, *the following limit holds*

$$
\sup_{p \in \mathcal{D}_d} P\left( \|V_n - p\|_1 > 3 \min_{q \in \mathcal{H}_{d,b}^k} \|p - q\|_1 + \varepsilon \right) \to 0,
$$

*where* $V_n$ *is a function of* $X_1, \ldots, X_n \overset{iid}{\sim} p$.

Likewise the rate on $bk + k^d$ can also not be significantly improved in Theorem 2.2.

**Theorem 2.4.** *Let* $d \geq 2$, $b \to \infty$, *and* $k \to \infty$ *with* $b \geq k$ *and* $n/(bk + k^d) \to 0$. *There exists no estimator* $V_n \in \widetilde{\mathcal{H}}_{d,b}^k$ *such that, for all* $\varepsilon > 0$, *the following limit holds*

$$
\sup_{p \in \mathcal{D}_d} P\left( \|V_n - p\|_1 > 3 \min_{q \in \widetilde{\mathcal{H}}_{d,b}^k} \|p - q\|_1 + \varepsilon \right) \to 0,
$$

*where* $V_n$ *is a function of* $X_1, \ldots, X_n \overset{iid}{\sim} p$.

### 2.2.2 Finite-Sample and Distribution-Independent Bounds

In this section we investigate the convergence of multi-view histogram estimators to densities that satisfy the multi-view assumption. This will be done via a standard bias/variance decomposition style of argument. We begin with the following distribution-dependent finite-sample bound.

**Proposition 2.1.** *Let* $d, b, k, n \in \mathbb{N}$ *and* $0 < \delta \leq 1$. *There exists an estimator* $V_n \in \mathcal{H}_{d,b}^k$ *such that*

$$
\sup_{p \in \mathcal{D}_d} P\left( \|p - V_n\|_1 > 3 \min_{q \in \mathcal{H}_{d,b}^k} \|p - q\|_1 + 7\sqrt{\frac{2bdk \log(4bdkn)}{n}} + 7\sqrt{\frac{\log(\frac{3}{\delta})}{2n}} \right) < \delta,
$$

*where* $V_n$ *is a function of* $X_1, \ldots, X_n \overset{iid}{\sim} p$.

To analyze the approximation error, $\|p - q\|_1$, we first consider the case where $p$ has a single component with $L$-Lipschitz marginals, so $p = \prod_{i=1}^{d} p_i$. Using the indicator function over the unit cube with Hölder's Inequality we have that $\|p - q\|_1 = \|(p - q)\mathbb{1}\|_1 \leq \|p - q\|_2 \|\mathbb{1}\|_2 = \|p - q\|_2$. It is possible to show that the $L^2$ projection of $p$ onto $\mathcal{H}_{d,b}^1$ is achieved by simply projecting each marginal to its best approximating one-dimensional marginal i.e. (see the appendix)

$$
\arg\min_{q \in \mathcal{H}_{d,b}^1} \left\| \prod_{i=1}^{d} p_i - q \right\|_2 = \prod_{i=1}^{d} \mathrm{Proj}_{\mathcal{H}_{1,b}} p_i.
$$

Let $\mathrm{Lip}_L$ be the set of $L$-Lipschitz probability density functions $[0, 1]$ and let $m_L \triangleq \sup_{f \in \mathrm{Lip}_L} \|f\|_2$. The following theorem characterizes the approximation error for separable densities with Lipschitz continuous marginals.

**Theorem 2.5.** *Let $b^2 \geq L^2/12$ and $f_1, \ldots, f_d$ be elements of $\mathrm{Lip}_L$ then*

$$\left\| \prod_{i=1}^d f_i - \mathrm{Proj}_{\mathcal{H}_{d,b}^1} \prod_{i=1}^d f_i \right\|_1 \leq \sqrt{m_L^{2d} - \left( m_L^2 - \frac{L^2}{12b^2} \right)^d}.$$

We show in the appendix that this decays at a rate of $O\left(b^{-1}\right)$ and that this rate is tight. To characterize the behavior of $m_L$ we have the following.

**Proposition 2.2.** *Let $m_L = \sup_{f \in \mathrm{Lip}_L} \|f\|_2$, then*

$$m_L^2 = \begin{cases} L^2/12 + 1 & 0 \leq L \leq 2 \\ \sqrt{\frac{8L}{9}} & L \geq 2 \end{cases}.$$

The following theorem is a finite-sample bound for a well-constructed multi-view histogram estimator.

**Theorem 2.6.** *Fix $n, d, k \in \mathbb{N}$, $L \geq 2$, and $1 \geq \delta > 0$. There exists $b$ and an estimator $V_n \in \mathcal{H}_{d,b}^k$ such that*

$$\sup_{p \in Q} P\left( \|V_n - p\|_1 > \frac{21dk^{1/3}L^{\frac{d+3}{12}}}{n^{\frac{1}{3}}} \sqrt{\log(3Ldkn)} + 7\sqrt{\frac{\log(\frac{3}{\delta})}{2n}} \right) < \delta,$$

*where $Q$ is the set of densities of the form $\sum_{i=1}^k w_i \prod_{j=1}^d p_{i,j}$ with $p_{i,j} \in \mathrm{Lip}_L$, $w \in \Delta_k$, and $V_n$ is a function of $X_1, \ldots, X_n \overset{iid}{\sim} p$.*

Here we analyze the asymptotic rate at which the estimator in Theorem 2.6 converges to a large class of multi-view models. To this end let

$$\mathfrak{Q}_d = \left\{ \sum_{i=1}^{k'} w_i p_{i,1} \otimes \cdots \otimes p_{i,d} \mid k' \in \mathbb{N}, w \in \Delta_{k'}, L' \geq 0, p_{i,j} \in \mathrm{Lip}_{L'} \right\},$$

i.e. $\mathfrak{Q}_d$ is the space of all multi-view models whose component marginals are all Lipschitz continuous. Consider letting $L \to \infty$, $k \to \infty$, and $\delta \to 0$ in Theorem 2.6 arbitrarily slowly as $n \to \infty$. For some element $p$ in $\mathfrak{Q}_d$ its respective maximum Lipschitz constant and component number, $L'$ and $k'$, are fixed, so for sufficiently large $n$ we have $L > L'$ and $k > k'$ and the bound from Theorem 2.6 applies. From this it follows that $\|V_n - p\|_1 \in \widetilde{O}\left(1/\sqrt[3]{n}\right)$. So we can construct an estimator that converges at rate $\widetilde{O}\left(1/\sqrt[3]{n}\right)$ for *any* multi-view model in $\mathfrak{Q}_d$, independent of dimension! This rate appears to be approximately optimal since "for smooth densities, the average $L^1$ error for the histogram estimate must vary at least as $n^{-1/3}$" ([14], p. 99). For comparison the histogram estimator's convergence rate is hindered exponentially in dimension.

**Proposition 2.3.** *Let $V_n$ be the standard histogram estimator with $n/b^d \to \infty$. There exists $p \in \mathfrak{Q}_d$ such that $\|V_n - p\|_1 \in \omega(1/\sqrt[d]{n})$.*

## 2.3 Discussion

While Theorem 2.6 gives an estimator with good convergence, the class of densities $Q$ is somewhat restrictive and likely not realistic for many situations where one would want to apply a nonparametric density estimator. Proposition 2.1, on the other hand, is not so restrictive since it depends on $\min_{q \in \mathcal{H}_{d,b}^k} \|p - q\|_1$, and more clearly conveys the message of this paper. Typical works on multi-view nonparametric density estimation assume that the target density $p$ is a multi-view model and are interested in recovering the model components $p_{i,j}$ and $w$ from (1). Similarly to how the standard histogram estimator doesn't assume $p \in \mathcal{H}_{d,b}$ our work isn't meant to assume that $p$ is a multi-view model, but is instead meant to explore the benefits of including a hyperparameter $k$, in addition to $b$, to restrict rank of the estimator. Just as the histogram can approximate any density as $b \to \infty$, Lemmas 2.1 and 2.2 show that the inclusion of $k$ does not hinder the approximation power of the estimator.

To explore the trade-off between $b$ and $k$ first observe that setting $k = b^d$ gives $\mathcal{H}_{d,b}^k = \mathcal{H}_{d,b}$ since one can allocate one component to each bin. Theorem 2.1 and Proposition 2.1 with $k = b^d$ gives a sample complexity of approximately $n \sim b^{d+1}$ which coincides with standard histogram estimator. Alternatively, setting $k = 1$ restricts the estimator to separable histograms $\mathcal{H}_{d,b}^1$, with a sample complexity of approximately $n \sim b$. Thus we have a span of $k$ yielding different estimators with maximal $k$ corresponding to the standard histogram and minimal $k$ corresponding to a naive Bayes assumption. We observe in Section 3 that this trade-off is useful in practice: we virtually never want $k$ to be maximized.

## 3 Experiments

The previous section proved the existence of estimators that select NNTF histograms that offer advantages over the standard histogram. Unfortunately these estimators are not computationally tractable and only demonstrate the potential benefit incorporating an NNTF model in nonparametric density estimation. It would be nonetheless interesting to observe the behavior of an NNTF histogram estimator, even if it lacks the theoretical guarantees developed in the previous section. To this end we consider an $L^2$-minimizing NNTF histogram estimator. For all $d, b, k$ fix $\mathcal{R}_{d,b}^k$ to be either $\mathcal{H}_{d,b}^k$ or $\widetilde{\mathcal{H}}_{d,b}^k$. We consider an estimator $U_n$, that attempts to minimize

$$U_n = \mathrm{argmin}_{\hat{p} \in \mathcal{R}_{d,b}^k} \|\hat{p} - p\|_2^2.$$

Note that

$$\|\hat{p} - p\|_2^2 = \|\hat{p}\|_2^2 - 2 \langle p, \hat{p} \rangle + \|p\|_2^2. \tag{5}$$

The $\|p\|_2^2$ term in (5) is not relevant when minimizing over $\hat{p}$. Additionally for data $X_1, \ldots, X_n \overset{iid}{\sim} p$ the law of large numbers gives us

$$\langle p, \hat{p} \rangle = \int_{[0,1)^d} p(x)\hat{p}(x)dx = \mathbb{E}_{X \sim p}[\hat{p}(X)] \approx \frac{1}{n} \sum_{i=1}^{n} \hat{p}(X_i)$$

so we may consider

$$U_n \triangleq \mathrm{argmin}_{\hat{p} \in \mathcal{R}d,b} \|\hat{p}\|_2^2 - \frac{2}{n} \sum_{i=1}^{n} \hat{p}(X_i) \tag{6}$$

as a practical version of (5) which is conveniently equivalent to nonnegative tensor factorization (see the appendix).

Risk expressions for nonparametric density estimation based on $L^2$-minimization similar to (6) have appeared in previous works related to kernel density estimation [41, 8, 31]. Proving optimal rates on bandwidth in these settings seems challenging, however.

Table 1: Experimental Results

| Dataset | $d$ Red. | Dim. | Hist. Perf. | Tucker Perf. | Hist. Bins | Tucker Bins | Tucker $k$ | $p$-val. |
|---------|----------|------|-------------|--------------|------------|-------------|------------|----------|
| MNIST | PCA | 2 | -1.455±0.089 | -1.502±0.102 | 6.531±1.499 | 8.375±1.780 | 4.968±1.976 | 5e-4 |
| | | 3 | -2.040±0.196 | -2.268±0.195 | 4.781±0.738 | 6.718±1.565 | 5.781±1.340 | 2e-4 |
| | | 4 | -3.532±0.996 | -4.014±0.655 | 4.031±0.585 | 5.343±1.018 | 4.375±0.695 | 2e-3 |
| | | 5 | -4.673±1.026 | -6.157±2.924 | 3.468±0.499 | 4.343±0.592 | 3.281±0.514 | 4e-5 |
| | Rand. | 2 | -2.034±0.100 | -2.099±0.102 | 6.062±1.197 | 7.562±1.657 | 2.062±1.784 | 3e-5 |
| | | 3 | -3.086±0.207 | -3.331±0.387 | 4.812±0.526 | 6.843±1.227 | 2.687±1.959 | 1e-4 |
| | | 4 | -4.307±0.290 | -5.731±0.435 | 3.500±0.585 | 5.656±0.642 | 2.593±1.497 | 8e-7 |
| | | 5 | -6.327±0.522 | -9.539±1.053 | 3.250±0.433 | 4.718±0.571 | 2.562±1.087 | 8e-7 |
| Diabetes | PCA | 2 | -2.079±0.122 | -2.212±0.132 | 5.718±1.304 | 7.468±1.478 | 1.062±0.242 | 8e-6 |
| | | 3 | -3.010±0.364 | -3.606±0.420 | 3.593±0.860 | 7.062±1.058 | 1.843±1.543 | 2e-6 |
| | | 4 | -4.002±0.415 | -4.423±0.701 | 3.000±0.000 | 5.906±0.804 | 2.343±1.107 | 2e-3 |
| | | 5 | -6.139±0.661 | -6.043±1.192 | 3.000±0.000 | 3.750±0.968 | 1.843±0.617 | 0.91 |
| | Rand. | 2 | -3.074±0.224 | -3.277±0.287 | 6.843±1.227 | 9.250±1.936 | 1.093±0.384 | 7e-5 |
| | | 3 | -4.726±0.483 | -5.353±0.751 | 4.968±0.769 | 8.406±1.343 | 1.625±1.672 | 2e-5 |
| | | 4 | -6.017±0.873 | -7.732±1.497 | 4.062±0.704 | 6.718±1.328 | 2.093±1.155 | 1e-5 |
| | | 5 | -8.986±1.292 | -12.61±2.477 | 3.062±0.242 | 5.093±0.521 | 2.531±0.865 | 2e-6 |

For our experiments we used the Tensorly library [23] to perform the nonnegative Tucker decomposition [22] with Tucker rank $[k, k, \ldots, k]$ which was then projected to the simplex of probability

tensors using [12]. We also performed experiments with nonnegative PARAFAC decompositions using [34, 23]. These decompositions performed poorly. This is potentially because the PARAFAC optimization is more difficult or the additional flexibility of the Tucker decomposition was more appropriate for the experimental datasets.

## 3.1 Experimental Setup

Our experiments were performed on the Scikit-learn [28] datasets MNIST and Diabetes [28], with labels removed. We use the estimated risk from (6) to evaluate estimator performance (which may cause negative performance values, however lower values always indicate smaller estimated $L^2$-distance). Our experiments considered estimating histograms in $d = 2, 3, 4, 5$ dimensional space. We consider two forms of dimensionality reduction. First we consider projecting the dataset onto its top $d$ principle components. We also performed experiments projecting each dataset onto a random subspace of dimension $d$. These random subspaces were constructed so that each additional dimension adds a new index without affecting the others. To do this, we first randomly select an orthonormal basis for each dataset that remains unchanged for all experiments $v_1, v_2, \ldots$. Then to transform a point $X$ to dimension $d$ we perform the following transform: $X_{\text{reduced dim.}} = [v_1 \cdots v_d]^T X$. We consider both transforms since PCA may select dimensions where the features tend to be independent, e.g. a multivariate Gaussian. After dimensionality reduction we scale and translate the data to fit in the unit cube.

With our preprocessed dataset, each experiment consisted of randomly selecting 200 samples for training and using the rest to evaluate performance (again using (6)). For the estimators we tested all combinations using 1 to $b_{\max}$ bins per dimension and $k$ from 1 to $k_{\max}$. As $d$ increased the best cross validated $b$ and $k$ value decreased, so we reduced $b_{\max}$ and $k_{\max}$ for larger $d$ to reduce computational time, while still leaving a sizable gap between the best cross validated $b$ and $k$ and $b_{\max}$ and $k_{\max}$ across all runs of all experiment. For $d = 2, 3$ we have $b_{\max} = 15$ and $k_{\max} = 10$; for $d = 4$ we have $b_{\max} = 12$ and $k_{\max} = 8$; for $d = 5$ we have $b_{\max} = 8$ and $k_{\max} = 6$. For parameter fitting we used random subset cross validation repeated 80 times using 40 of the 200 samples to evaluate the performance of the estimator fit using the other 160 samples. Performing 80 folds of cross validation was necessary because of the high variance of the histogram's estimated risk. This high variance is likely due to the noncontinuous nature of the histogram estimator itself and the noncontinuity of the histogram as a function of the data, i.e. slightly moving one training sample can potentially change histogram bin in which it lies. Each experiment was run 32 times and we report the mean and standard deviations of estimator performance as well as the best parameters found from cross validation. We additionally apply the two-tailed Wilcoxon signed rank test to the 32 pairs of performance results to statistically determine if the mean performance between the standard histogram and our algorithm are different and report the corresponding $p$-value.

## 3.2 Results

Our results are presented in Table 1. Apart from the density estimators' performance ("Hist. Perf." and "Tucker Perf.") this table contains the mean and standard deviation over the 32 trials for the optimal cross validated parameters for the estimators. This includes the number of bins, corresponding to $b$ from the earlier sections, and the Tucker rank $k$. We see that the Tucker histogram usually outperforms the standard histogram ("Tucker Perf." is less than "Hist. Perf.") with high statistical significance (small $p$-val). As one would expect, the Tucker histogram cross validates for more bins, presumably reducing the $k$ in exchange for more bins. Note that the MNIST PCA experiments always cross validated for the largest $k$ and yielded less statistically significant performance increases (outside of the unusual Diabetes PCA $d = 5$ experiment). Presumably this experiment fit the NNTF assumption the least well (hence the large $k$) and thus received the least benefit.

## 4 Conclusion

In this paper we have theoretically and experimentally demonstrated the advantages of including rank restriction into nonparametric density estimation. In particular, rank restriction may be a way to overcome the curse of dimensionality since it reduces the sample complexity penalty incurred from dimensionality from exponential to linear. This paper is an initial theoretical foray for demonstrating the potential of combining rank restriction with nonparametric methods.

## Acknowledgments and Disclosure of Funding

Robert A. Vandermeulen acknowledges support by the Berlin Institute for the Foundations of Learning and Data (BIFOLD) sponsored by the German Federal Ministry of Education and Research (BMBF). Antoine Ledent acknowledges support by the German Research Foundation (DFG) award KL 2698/5-1 and the Federal Ministry of Science and Education (BMBF) award 01IS18051A. The authors thank Marius Kloft and Clayton Scott for useful discussions related to this work. The authors thank Nicholas D. Sidiropoulos for bringing to our attention some previous works on low-rank nonparametric density estimation.

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
