# OpenReview forum: "Beyond Smoothness: Incorporating Low-Rank Analysis into Nonparametric Density Estimation"
_NeurIPS.cc/2021/Conference — NeurIPS 2021 Poster_

### Official Review · Reviewer_JhG3 · 2021-07-14

**Rating:** 6
**Confidence:** 2

**Summary:**

This work is concerned with the (high-dimensional) density estimation problem.  While most minimax rates  for  the  problem  are  expressed  in  terms  of  some  measure  of  smoothness  that  the  unknown  distribution is assumed to verify, the current approach adds more structure, by requiring that the joint density be tensor-decomposable (in a PARAFAC or in a Tucker sense), with Lipschitz components.  The authors show(theoretical analysis) that this subverts the smoothness-only minimax rates together with their curse of dimensionality. In order to illustrate their method, the authors further perform experiments in which –perhaps controversially– they relax the estimator search to an L2 problem.

**Ethical Concerns:**

None.

**Limitations And Societal Impact:**

Adequately addressed.

**Main Review:**

The results seem to be novel. The value of this work is primarily of a theoretical nature, as the estimators are not computationally tractable (as made clear by the authors). This reviewer did not very correctness of the proofs in the appendix.  The authors have addressed the comments that reviewers have raised while under review for previous conferences. Slightly tends towards acceptance.

* On p.2,  the authors already mentions the estimators are not computationally tractable. Briefly already state why here (or at least refer to the Main Technical Tools paragraph).

* Could the authors comment and provide the reader with some intuition about the obtained rate
($1/\sqrt[3]{n}$ instead of $1/\sqrt[d]{n}$). Why should we expect a cubic root?

* What would happen if we were to consider components with different degrees of smooth-ness (more general Holder classes) instead of Lipschitz?

* As a follow-up question:  do you expect the obtained rates to be tight;  namely, can one construct a corresponding lower bound?

* Further explain what the different columns of Table 1 correspond to.

* Typos.
    * p.1  nonparametric
    * p.3  asymptotically
    * p.4  entries
    * p.7  Lipschitz

**Time Spent Reviewing:**

3

---

> ### Author Response · Authors · 2021-08-10
> **Reviewer JhG3 Response**
>
> Thank you for your review, we will incorporate your corrections into our revision.
> Addressing your points in order:
>
>
> __Computational tractability__: The estimators in our theorems operate by constructing an epsilon net over the space of densities under consideration (in this case low rank histograms) and then performing many pairwise comparisons using the available data to select the best histogram. If k is the cardinality of the net then the computational complexity is worse than $k^2$. More recent works have reduced this to linear time (Density estimation in linear time, Mahalanabis and Stefankovic COLT 2008), but the covering numbers remain quite large.
>
> __Convergence rate__: Put simply, the $n^{-1/3}$ rate is what we were able to achieve with the tools we collected. While looking into answering this question we found that $n^{-1/3}$ is a lower bound for the $L^1$ rate of convergence of histograms in one dimension: “for smooth densities, the average $L^1$ error for the histogram estimate must vary at least as $n^{-1/3}$” p.99 Nonparametric Density Estimation: The $L^1$ View, Devroye and Györfi 1985. So our rate is approximately optimal, at least for fixed k.
>
> __Other smoothness characterizations__: We expect a similar rate of convergence. The estimators we are analyzing are quite good at selecting a nearly optimal estimator from a large collection of candidate densities, so this analysis depends solely on the convergence of the best approximating histogram as the bin width decreases and rank increases. One could alternatively integrate this assumption into the estimator which would reduce the space candidate densities greatly and could be an interesting direction for future research.
>
> __Lower bound__: See above.
>
> __Table columns__: These correspond to the datasets, the dimensionality reduction method, the number of dimensions kept in the dimensionality reduction method, the performance of the standard histogram estimator, the performance of the low rank estimator (smaller is better for these, more details are in the appendix), the cross validated parameters for the two density estimation methods and the p-value associated with the null hypothesis that the standard histogram’s performance is the same on average as the low rank histogram. Thank you for pointing this out and we will clarify this in our revision.

---

> > ### Comment · Reviewer_JhG3 · 2021-08-31
> > **Thank you**
> >
> > Many thanks for the detailed response.
> > The reviewer is happy that the comments lead the authors to consider optimality concerns,
> > and recommends to increase this discussion about rates and lower bounds.
> > This will lead to a stronger submission.

---

### Official Review · Reviewer_W5MH · 2021-07-16

**Rating:** 5
**Confidence:** 3

**Summary:**

The authors propose and analyse a novel method for nonparametric density estimation. They provide a method for providing a density estimate with low rank structure which is a mixture of densities that factorizes across dimensions, with each factor defined as a one dimensional histogram. They provide conditions under which their estimator is consistent and characterize upper and lower bounds on its error.

**Limitations And Societal Impact:**

One sentence describing no foreseen societal impacts is provided.

**Main Review:**

The approach described is novel, and the theoretical results are interesting and nontrivial.

The paper has some limitations I will highlight.  The authors could improve the paper by giving more context into why they are solving the problem they address.  In particular, given that the improvement obtained relative to standard histograms is most pronounced in high dimensions, in what settings might it be useful to have a 3 dimensional or higher density estimate of this form?  When the number of samples is large, does this matter for downstream analyses?  If so, of what sort of downstream analyses.

The paper should be much better polished.  The paper has many typos, and missing information -- some of this impedes understanding.
“Nonparametreic” in the second sentence.
In section 1.3, what is “M” (i.e. “M satisfies a property”).
“Who’s” should be “whose”
“One can recover y”, should “y” be “x”?

In section 1.3, what does that last sentence of the second to last paragraph mean?  What is “our setting”, and why is this an important setting to consider?

What is table 1?  This is not referenced in the main text (or maybe “table 3” is a typo?), and the legend “Experimental results” is not informative.  Are higher or lower values better?

Some sentences do not make sense grammatically. E.g. in section 3, “This problem when seems be further…”

The authors have not included line numbers, which would have been helpful for referencing certain parts of the paper.

**Time Spent Reviewing:**

4

---

> ### Author Response · Authors · 2021-08-10
> **Reviewer W5MH Response**
>
> __Context__: We don’t fully understand your question. Particularly “...what settings might it be useful to have a 3 dimensional or higher density estimate of this form? When the number of samples is large, does this matter for downstream analyses? If so, of what sort of downstream analyses.”
>
> We are proposing our approach as a general means of improving density estimation which is a well-established problem (frequently its own subject area in conference submissions) oftentimes involving data of 3 or more dimensions. Its difficult to speak concretely about potential benefits to “downstream analyses” since the application space of density estimation is rather vast. But, if the downstream analysis would benefit from a more accurate density estimator, then our results seem to imply that they would benefit from using a low-rank approach.
>
> For some context to the general concept behind our idea, our approach was inspired by matrix completion where it was found that a low-rank assumption is _generally_ useful and matrices with slow spectral decay are rather unusual (Spectral Regularization Algorithms for Learning Large Incomplete Matrices, R. Mazumder et al. JMLR 2011). In this paper we demonstrate that this assumption is also useful for density estimation and precisely characterize this benefit mathematically. We address densities that are pathologically bad in terms of low-rank approximations in Appendix E. We hope this addresses your concern.
>
>
> __Polishing__: Thank you for your corrections, we will incorporate them into further revisions. We intend to continue polishing the paper until the camera-ready submission.

---

> > ### Comment · Reviewer_W5MH · 2021-08-15
> > **Response to Rebuttal**
> >
> > Thank you for your reply.
> >
> > My question boils down to: What is the significance of your theoretical results?
> >
> > I consider myself to be an expert in theoretical machine learning broadly, but not an expert in density estimation (I having written only one paper on the topic).  However, as currently written, the paper does not help me understand how / if the contribution could be significant to the broader NeurIPS community.  At present, I can say only that the methods presented are mathematically interesting and enjoyable to read.
> >
> > My concern is that as far as I know, >3 dimensional density estimates are never of immediate interest to practitioners working with real data, who are not machine learning researchers; density estimates are often useful, for example, as a route to estimating things like moments and quantiles.  If you disagree (or better yet, have a counter example), I feel that arguing this would dramatically strengthen the paper.
> >
> > I would be happier to attest to the significance of the contribution if you describe how the theoretical results of the paper might (even hypothetically) improve over a simple histogram by (for example) 1.  Enabling someone to learn something new with less data (e.g. via a hypothesis test),  2. Providing a better prediction of some sort, or 3. Doing something else concrete a practitioner using your method might want to do.   Or if it is not clear whether the contribution is useful on its own, but it might help provide a better foundation upon which other practical methods for high dimensional density estimation might build, that can be fine too if it is articulated in the paper.

---

> > > ### Author Response · Authors · 2021-08-17
> > > **More context with applications**
> > >
> > > Thank you for clarifying this for us. The following are three examples of the use of higher dimensional nonparametric density estimation which support the significance of higher dimensional nonparametric density estimation. Links to relevant articles are provided after the main text.
> > >
> > > __Monte Carlo Simulation__: Density estimation is often used for Monte Carlo simulations with systems that use a particle density $\phi$ in phase space. Phase space can be up to 7 dimensions (location$\times 3$, velocity$\times 3$, time) and such densities are estimated using samples from expensive particle simulations. The following PhD thesis is dedicated to this subject and mentions histogram models specifically: _Kernel Density Estimation Techniques for Monte Carlo Reactor Analysis_.
> > >
> > > __Anomaly Detection__: A common approach to anomaly detection is to fit the data with a density estimator with low likelihood samples being declared anomalous. A more accurate density estimator here results in improved false alarm and anomaly miss rates.
> > >
> > > Using a nonparametric density estimator for these problems is common. _A Unifying Review of Deep and Shallow Anomaly Detection_, a recent article in The Proceedings of the IEEE, states that “The standard KDE [kernel density estimator]... is therefore a popular approach to AD [anomaly detection].” The review also mentions histograms as an approach to anomaly detection.
> > >
> > > For a concrete application of histograms to anomaly detection involving distributions with tens of dimensions we refer to _Histogram-Based Traffic Anomaly Detection_.
> > >
> > > __Entropy Estimation__: Nonparametric density estimation is often used as an intermediary to estimating density functionals such as entropy. In the early tract _On The Estimation of Entropy_ we have the following:
> > >
> > > “We point out that histogram estimators may be used to construct root-n consistent entropy estimators in p = 1 dimension, and that kernel estimators give root-n consistent entropy estimators in p = 1, 2 and 3 dimensions, but that neither type generally provides root-n consistent estimation beyond this range, unless (for example) the underlying distribution is compactly supported, or is particularly smooth and bias-reduction techniques are employed.”
> > >
> > > Our method fits nicely into a “bias-reduction” technique, since it assumes a specific structure (low-rank) on the underlying density.
> > >
> > > More recent works, such as the NeurIPS paper _Ensemble weighted kernel estimators for multivariate entropy estimation_, provide rates on entropy estimators. However, those rates still exhibit poor convergence properties, reflecting the poor rate (in dimensionality) we find for the standard histogram density estimator in our paper: “A large class of estimators of functionals of the probability density suffer from the curse of dimensionality, wherein the mean squared error decays increasingly slowly as a function of the sample size as the dimension of the samples increases. In particular, the rate is often glacially slow of order  $O(T^{-\gamma/d})$ , where $\gamma$ is a rate parameter.”
> > >
> > > Again our method may provide a means for estimating such functionals, especially in higher dimension regimes.
> > >
> > >
> > > We hope we have addressed your concerns. We are happy to address further questions and concerns. Later revisions will include more discussion of such applications.
> > >
> > > Links to relevant documents:
> > >
> > >
> > > Kernel Density Estimation Techniques for Monte Carlo Reactor Analysis https://deepblue.lib.umich.edu/bitstream/handle/2027.42/135841/tpburke_1.pdf
> > >
> > > A Unifying Review of Deep and Shallow Anomaly Detection
> > > https://ieeexplore.ieee.org/document/9347460
> > >
> > > Histogram-Based Traffic Anomaly Detection
> > > https://ieeexplore.ieee.org/abstract/document/5374831
> > >
> > > On The Estimation of Entropy
> > > https://link.springer.com/content/pdf/10.1007/BF00773669.pdf
> > >
> > > Ensemble weighted kernel estimators for multivariate entropy estimation
> > > https://proceedings.neurips.cc/paper/2012/hash/92c8c96e4c37100777c7190b76d28233-Abstract.html

---

> > > > ### Comment · Reviewer_W5MH · 2021-08-17
> > > > **Reply to context and applications**
> > > >
> > > > I appreciate this context and look forward to seeing this discussion in a later revision.  It will be nice as well to see how your theoretical results tie into performance in these applications, for example, on how the convergence rates you have proved relate to relate to convergence rates for entropy estimates.
> > > >
> > > > I will raise my score by one point, in expectation of your inclusion of some of this discussion of applications and limitations of existing methods in these contexts in your paper.

---

### Official Review · Reviewer_gXvQ · 2021-07-16

**Rating:** 5
**Confidence:** 4

**Summary:**

The paper shows a number of results on non-parametric density estimation using mixtures of product histograms with two proposed methods called \textit{multi-view histograms} and \textit{Tucker histograms}. A simple experiment is also performed to show these methods work better than the usual histogram estimation in some specific cases. There are two major theoretical contributions as follows.  Firstly, the authors give some asymptotic error control for the proposed methods for any true density functions based on sample sizes, number of histograms' bins, and number of mixtures.  Secondly, they prove that when the true density is a mixture of product distributions, the proposed methods are more favorable compared to the traditional histograms method in terms of the convergence rate.

**Ethical Concerns:**

I do not detect any ethical issues with the paper.

**Ethics Review Area:**

["I don’t know"]

**Limitations And Societal Impact:**

I do not see any foreseeable limitations and negative societal impact of this work.

**Main Review:**

--- Strength:

(1) The theoretical problems above are well studied in the paper. All the results are rigorously presented.

(2) The techniques using here are well-known and are applied appropriately to support the proofs. The assumptions of the problem are reasonable and useful.

(3) The experiment also sheds some light on the superiority of the proposed method in some specific cases.

--- Weakness:

(1) The title: "Beyond smoothness": I can not see what is beyond the smoothness here as we still need to assume the smoothness of true density to prove all the results. Even in the case that the true density is "low-rank", the assumptions on smoothness of product densities are required.

(2) Experiments: The shown experiment is too rudiment and does not strongly support the argument in theory part. The description of the experiment is too detailed while the discussion and explanation parts are short.

--- Point to improve:

(1) In the experiment part, the authors can conduct an simulation study to show the rate when approximating by multi-view histograms compared to the usual histograms method.

(2) We can also find a number of real data sets having mixtures of product distributions law and try to compare the proposed methods on them. They appear a lot in psychological field (\url{https://arxiv.org/pdf/1904.04378.pdf}), see also the papers which cite \url{https://projecteuclid.org/journals/annals-of-statistics/volume-37/issue-6A/Identifiability-of-parameters-in-latent-structure-models-with-many-observed/10.1214/09-AOS689.full?tab=ArticleLinkCited} for more information.

--- References:

The authors may consider adding some recent references on multivariate density estimation with very fast MISE via sin kernel, such as via the Fourier integral theorem [1]. In particular, this work shows that when the density function has supersmooth Fourier tails, such as Gaussian distribution, the MISE rate is $n^{-1/2}$ (up to some logarithmic term). Furthermore, the usage of sin kernel also circumvents challenge with choosing covariance matrix in standard Gaussian kernel while it is able to learn the dependency in data automatically.

[1] N. Ho, S. G. Walker. Multivariate smoothing via the Fourier integral theorem and Fourier kernel. Arxiv preprint Arxiv: 2012.14482, 2021.

**Time Spent Reviewing:**

4 hours

---

> ### Author Response · Authors · 2021-08-10
> **Reviewer gXvQ Response**
>
> Thank you for your review. The included references seem helpful and we will keep them in consideration for further works. We will address your “weaknesses” points in order.
>
> __Title__: We did not mean to imply that “smoothness” is totally eliminated from our analysis but that one can consider model properties other than (“beyond”) smoothness, in this case low-rank structure. To produce a finite sample characterization of the convergence of the estimator to the true density we do indeed need a characterization of the smoothness of the true density; we are unaware of any approach to this that does not make a similar assumption (either Lipschitz continuous, in a Hölder or Sobolev class of functions, or some similar class of functions) since one can choose the true density to be arbitrarily badly approximated by a given function class. However we do show that estimation error (disregarding bias) (Prop 2.1) and universal consistency (Lemmas 2.1 and 2.2) do not depend on smoothness.
>
> __Experiments__: We agree with you in regards to the discussion of the experimental results. There are some rather interesting phenomena regarding the experiments which we removed due to lack of space and to fit with suggestions from our last review. We will defer more of the experimental description to the appendix and include the following points in the discussion:
>
> * We see that the Diabetes dataset is behaving somewhat strangely on the PCA. Interestingly, a previous reviewer suggested we add the random subspace experiments due to PCA potentially selecting for independence between subspaces, but it seems that typicality (total randomness) tends to select for subspaces that are more amenable to our method.
> * We see that the highest dimensionality tends to exhibit the smallest p values suggesting that our method does indeed see the greatest benefit at higher dimensionalities.
> * The low-rank method always cross validates for more bins than the standard histogram, thereby exhibiting the benefit we would hope to see with a low-rank method.
>
> In regards to the following point:
> >(2) The techniques using here are well-known and are applied appropriately to support the proofs.
>
> We would like to mention that we use some rather novel analysis in the proof of Theorem 2.5. In particular Appendix C.1.1 is heavily inspired by work on a class of functional  transforms known as "monotonic rearrangements" (see "A Class of Monotone Decreasing Rearrangements" Epperson 1989) which we use to determine $L^2$ maximizing Lipschitz densities.

---

> > ### Comment · Reviewer_gXvQ · 2021-08-31
> > **Response to the authors**
> >
> > I would like the authors for your detailed response to my comments. I still find the experiment parts quite underwhelming and need to be greatly improved in future work. Furthermore, the novelty of the theoretical analysis part is still not convincing to me. Given these points, I still maintain my current score with the paper.

---

> > > ### Author Response · Authors · 2021-08-31
> > > **In regards to lack of novelty...**
> > >
> > > We would very much appreciate it if you would elaborate on the lack of novelty as this was not mentioned in your ”weaknesses” or “points to improve” and this concern stands in stark contrast to the other reviewers (as mentioned in the general response). We are aware of the following works on our topic, low rank nonparametric density estimation:
> > > * “Nonparametric Estimation of Multi-View Latent Variable Models” Le Song, Animashree Anandkumar, Bo Dai, and Bo Xie ICML 2014
> > >
> > > And the following three works by Magda Amiridi, Nikos Kargas, and Nicholas D. Sidiropoulos:
> > > * “Nonparametric Multivariate Density Estimation: A Low-Rank Characteristic Function Approach” arXiv 2020
> > > * “Low-rank Characteristic Tensor Density Estimation Part I: Foundations” arXiv 2021
> > > * “Low-rank Characteristic Tensor Density Estimation Part II: Compression and Latent Density Estimation” arXiv 2021
> > >
> > > However, none of these works were able to mathematically establish the _improved_ rate of convergence that is the core point of our work. Please let us know if we are missing some crucial piece of literature that renders our work not novel. From our viewpoint there is a burgeoning interest in the theory behind the topic we study, but no existing work succeeds in proving rates such as the ones we obtain, which makes our results quite novel and significant for this field.

---

### Official Review · Reviewer_LSLp · 2021-07-16

**Rating:** 7
**Confidence:** 4

**Summary:**

The authors consider the density estimation problem in multiple dimensions. They study the convergence properties of a low-rank estimator: the estimator is the sum of a few tensor products of univariate histograms. They give asymptotic and finite sample bounds for convergence. When the true density is a sum of k tensor products of Lipschitz densities, the authors show an $L_1$ convergence rate of $n^{-⅓}.$




**Ethical Concerns:**

No concerns.

**Limitations And Societal Impact:**

No concerns.

**Main Review:**

Theoretical results in the manuscript are substantial.

They only prove the existence of a low-rank estimator; they do not devise such an estimator.

It is surprising that there is no $d$ in the asymptotic convergence results. I did not have time to check the proof.

Minor comments, mostly typos:
- In the paragraph below equation (3), $x$ is called a matrix, even though it is a vector
- Page 5, just before "Main Technical Tools", should be $n^{-1/3}$ not $d^{-1/3}$
- Theorem 2.6: the first line in the statement has some indexing errors in the product
- Equation (12): sum should be until $n$, and a factor of $1/n$ is missing


**Time Spent Reviewing:**

6

---

> ### Author Response · Authors · 2021-08-10
> **Reviewer LSLp Response**
>
> Thank you for your review. We will incorporate your corrections into our final draft.

---

### Author Response · Authors · 2021-08-10
**General Review Response.**

We thank the reviewers for their reviews and are pleased that the main results of the paper were very well received:

Rev. LSLp: “Theoretical results in the manuscript are substantial.”

Rev. gXvQ: “The theoretical problems above are well studied in the paper. All the results are rigorously presented... The assumptions of the problem are reasonable and useful.”

Rev. W5MH: “The approach described is novel, and the theoretical results are interesting and nontrivial.”

Rev. JhG3: “The results seem to be novel.”

We plan on incorporating all the corrections found by the reviewers as well as further polishing the paper until the camera ready date.

Significant update for revision: While looking into a question posed by Rev. JhG3 we found that our $n^{-1/3}$ rate (omitting log terms) for fixed $k$ approximately matches an $L^1$ lower bound for one dimensional histograms thereby implying that our rate is approximately optimal as multidimensional histograms can be adapted to estimate one dimensional densities by simply fixing all but the first index of the samples:
“for smooth densities, the average $L^1$ error for the histogram estimate must vary at least as $n^{-1/3}$” (p.99 Nonparametric Density Estimation: The $L^1$ View, Devroye and Györfi). Here "smooth" means that the one dimensional density satisfies "(i) f absolutely continuous  with a.e. derivative f'; (ii) f' is bounded and continuous" (p.98) which is (essentially?) equivalent to a density being Lipschitz continuous.
We will include this in future revisions.

Below we further address the reviewers' points individually.

---

### Decision · Program_Chairs · 2021-09-27

**Decision:**

Accept (Poster)

**Comment:**

The authors present an analysis of two classes of histogram-based density estimators that incorporate low-rankness structures analogous to CP and Tucker tensor factorizations, and experiments show that in practice a simple heuristic motivated by the theory outperforms standard histograms. The major contribution of the paper is the theoretical analysis of the convergence of histogram estimators incorporating low-rankedness, both asymptotically and non-asymptotically, that shows their convergence rate is independent of the dimension. These results quantify the benefits of using low-ranked histogram estimators, as the convergence rate of the standard histogram estimator goes down with the dimension. They are significant contributions to the theoretical underpinnings of the growing are of low-rank density estimation.